# Humoral and Cell-Mediated Immunity Against SARS-CoV-2 in Healthcare Personnel Who Received Multiple mRNA Vaccines: A 4-Year Observational Study

**DOI:** 10.3390/idr17030042

**Published:** 2025-04-29

**Authors:** Hideaki Kato, Kaori Sano, Kei Miyakawa, Takayuki Kurosawa, Kazuo Horikawa, Yayoi Kimura, Atsushi Goto, Akihide Ryo

**Affiliations:** 1Infection Prevention and Control Department, Yokohama City University Hospital, Yokohama 236-0004, Japan; 2Emerging Infectious Diseases Research Center, Yokohama City University Hospital, Yokohama 236-0004, Japan; 3Influenza Research Center, National Institute of Infectious Diseases, Japan Institute for Health Security, Musashimurayama 208-0011, Japan; 4Clinical Laboratory Department, Yokohama City University Hospital, Yokohama 236-0004, Japan; 5Advanced Medical Research Center, Yokohama City University, Yokohama 236-0004, Japan; 6Department of Public Health, Yokohama City University School of Medicine, Yokohama 236-0004, Japan; 7Department of Bioinformatics and Integrative Omics, National Institute of Infectious Diseases, Japan Institute for Health Security, Musashimurayama 208-0011, Japan

**Keywords:** SARS-CoV-2, COVID-19, humoral immunity, updated vaccine, cell-mediated immunity, interferon-gamma releasing assay, 50% neutralizing titer, ELISpot, original antigenic sin, antibody divergence

## Abstract

**Background/Objectives:** The long-term effects of multiple updated vaccinations against severe acute respiratory syndrome coronavirus-2 (SARS-CoV-2) have not been clarified. Humoral or cellular immunity dynamics in healthcare workers for four years were analyzed. **Methods:** Blood samples were collected at five time points from April 2021 to January 2024. Humoral immunity was analyzed using the 50% neutralizing titer (NT_50_) against the original Omicron XBB and Omicron BA.2.86 strains and cellular immunity were analyzed using the ELISpot interferon-gamma releasing assay. NT_50_s and the spot-forming count (SFC) of the ELISpot assay were compared in the SARS-CoV-2 Omicron XBB-, Omicron-infected, and uninfected subjects. **Results**: 32 healthcare workers (median age, 47 years) who received 3–7 vaccine doses were enrolled. The NT_50_s against the original strain decreased after the second vaccination but were maintained after the third vaccine dose. NT_50_s against the Omicron XBB and BA.2.86 strains were detected before the Omicron vaccine was introduced and increased following the updated vaccination. The NT_50_s against the Omicron XBB and BA.2.86 strains were elevated after natural infection by the Omicron strain, albeit without differences compared with the findings in uninfected subjects. Multivariate regression analysis revealed no confounder that affected the antibody titer against the BA.2.86 strain at the fifth blood sampling. The median number of SFCs ranged from 78 to 208 after the first two doses. **Conclusions**: Multiple vaccinations induced the production of antibodies with divergent activity against emerging mutant strains and enhanced protective effects against the original strain. This finding supported the importance of updated vaccination.

## 1. Introduction

During the coronavirus disease 2019 (COVID-19) pandemic, humoral and cell-mediated immunity both protected against severe acute respiratory syndrome coronavirus-2 (SARS-CoV-2) infection and reduced the severity of natural infection. Antibody titers declined markedly 6 months after the initial vaccination, leading to the introduction of booster vaccines [1]. During the Omicron variant period, three doses of the mRNA vaccine were required to provide elemental immunity against SARS-CoV-2 [2]. Meanwhile, the vaccine’s effect in preventing severe disease was suggested to persist longer than its effect on antibody titers [3], suggesting that immune memory remains as cell-mediated immunity [4]. However, the longitudinal dynamics of humoral and cell-mediated immunity are not well understood. RNA viruses easily mutate, and SARS-CoV-2 evolved to spread widely [5,6]. Therefore, it is recommended that all individuals regularly receive updated vaccines [7]. In this background, we assessed humoral and cell-mediated immunity over 4 years in a cohort of healthcare workers who received multiple vaccinations, including subjects who were and were not infected by the SARS-CoV-2 Omicron strain. This study will provide data for clarifying the need for catch-up immunization with updated vaccines.

## 2. Materials and Methods

This study initially recruited 44 full-time healthcare workers at our hospital in April 2021. Ten subjects were excluded because fewer than four blood samples were obtained. Two additional subjects were excluded because they were infected with SARS-CoV-2 prior to receiving the first two doses of the vaccine. Finally, 32 participants who underwent four or five blood samplings between April 2021 and January 2024 were analyzed. As detailed in Figure 1, blood was collected from the participants at 6 weeks (time point A) and 6 months (time point B) after the first two vaccinations in 2021, immediately after the third vaccination and the emergence of the Omicron strain in January 2022 (time point C), after receipt of the Omicron BA.4/5 bivalent vaccine and the spread of the Omicron XBB strain in summer 2023 (time point D), and after receipt of the Omicron XBB monovalent vaccine and emergence of the Omicron JN.1 strain in January–February 2024 (time point E). Cell-mediated immunity was also assessed at time points B, C, and E.

Stored sera were brought to room temperature in February 2024 and assayed. The neutralizing titer assay was performed using an HIV-based pseudovirus bearing the SARS-CoV-2 spike [8]. Then, the 50% neutralizing titer (NT_50_) was calculated using ImageJ software version 1.54i (National Institute of Health, Bethesda, MD, USA). NT_50_ ≤ a20 was considered negative. When the serum exhibited no observable neutralizing activity to interpolate NT_50_, it was assigned an NT_50_ of 10. All samples were assayed in at least duplicate experiments. The original (D614G), XBB, and BA.2.86 strains were selected as targets for the NT_50_ assay because these strains were predominant in 2023 and 2024 and exhibited strong immune escape [8,9]. BA. 2.86 was the ancestral strain of the circulating variant in 2025.

To analyze cell-mediated immunity, the IFN-γ ELISpot assay (T-SPOT Discovery SARS-CoV-2, Oxford Immunotec Ltd., Abingdon, UK) was used. The IFN-γ ELISpot assay was performed by stimulating a SARS-CoV-2-specific peptide pool derived from the original strain [10]. IFN-γ is released by CD4^+^ and CD8^+^ T-lymphocytes; thus, the ELISpot assay likely measured responses from both cell types. The peptide pool for the spike protein used in the IFN-γ ELISpot assay excluded the S2 region (C-terminal) to avoid cross-reactivity with endemic coronaviruses [11]. The response to emerging variants did not differ considerably between the original strain and variants of concern, such as Delta and Omicron [12]. Blood samples for the ELISpot assay were submitted to the SRL (H.U. Holdings, Tokyo, Japan) on the same day of sampling. T-Cell Xtend reagent (Oxford Immunotec) was also used appropriately according to the manufacturer’s instructions. The number of spot-forming cells (SFCs) per 250,000 peripheral blood mononuclear cells (PBMCs) was calculated by subtracting the number of SFCs in the negative control panel from the number of SFCs in Spot 1 (peptide pool for spike protein). The number of SFCs was multiplied by four to express the final results per 1 × 10^6^ PBMCs.

Study subjects were queried regarding their vaccination history (number of vaccinations and their dates) and their COVID-19 infection history (dates of infection). During the study period, mRNA vaccines manufactured by Pfizer (New York, NY, USA) and Moderna (Cambridge, MA, USA) were available in Japan, and receipt of either vaccine was counted as vaccination. The health status of participants was also assessed. Subjects with a history of severe immunosuppression, including organ transplantation, hematopoietic stem cell transplantation, general oral corticosteroid therapy, hemodialysis, anticancer chemotherapy, and B-cell-depleting chemotherapy, were excluded.

The history of COVID-19 was defined as a laboratory-confirmed COVID-19 diagnosis or positivity for nucleocapsid protein IgG. The infecting variant was determined as the epidemiologically circulating variant at each time, as opposed to the RNA sequence.

Continuous data were reported as the mean and 95% confidence interval (CI), median and interquartile range (IQR), or geometric mean titer (GMT). Categorical data were presented as numbers and percentages. Data were analyzed using the Mann–Whitney U-test or Kruskal–Wallis test to compare continuous variables between two groups or among three groups, respectively. Multivariate regression analyses using potential confounders, including age, sex, the number of mRNA vaccine doses, and the SARS-CoV-2 infection history, were performed to identify the predictors of NT_50_. Receiver–operator analysis was used to predict NT50 values for distinguishing the history of SARS-CoV-2 infection. Statistical analyses were performed using Prism 10 (GraphPad Software, Boston, MA, USA). *p* < 0.05 was considered statistically significant. Written informed consent was obtained independently from each participant. This study was approved by the Yokohama City University Hospital ethics board (FF231100032).

## 3. Results

### 3.1. Subjects’ Characteristics

The median age of the participants in April 2021 was 47 years [42–51], and the cohort included 10 men and 22 women. The median number of mRNA vaccinations was six [5–7]. The time between the last vaccination and each blood collection time point was as follows: time point A, 6 ± 1 weeks; time point B, 176 days [174–180]; time point C, 23 days [21–28]; time point D, 25 days [18–160]; and time point E, 84 days [61–310]. In total, 32, 32, 31, 23, and 32 blood samples were obtained at time points A–E, respectively. Eighteen subjects had a history of infection during the Omicron period, whereas 14 had no history of SARS-CoV-2 infection. A summary of each participant’s vaccination and infection history, characteristics, and drawn blood samples is provided in the Appendix A.

### 3.2. Analysis of Humoral Immunity

The summary of the NT_50_ against each strain at each time point is presented in Figure 2. The GMTs of antibodies against the original strain (D614G) among the 32 subjects at time points A–E were 551.2, 57.9, 1801, 1676, and 1612, respectively (Figure 2A). The GMT for the D614G strain significantly decreased 6 months after the first two vaccinations (*p* < 0.001), but a rebound was noted after the third vaccination (no significant difference between time points C and D or between time points D and E). The GMTs of antibodies against the Omicron XBB and BA.2.86 strains were not detected at time points A and B, but they were detected at time point C, at which point the Omicron-contained vaccine had not been introduced. The GMTs of antibodies against the D614G strain were consistently higher than those against XBB.1.5 and BA.2.86 in the study subjects at time points C (1801, 22.0, and 38.9, respectively), D (1676, 100.4, and 132.7, respectively), and E (1612, 217.4, and 242.0, respectively).

At time point D, at which time the Omicron strain had emerged, subjects who had been infected with the Omicron strain exhibited higher NT_50_s against the XBB (GMT: 195.1 vs. 76.4, *p* = 0.053; Figure 2B) and BA.2.86 strains (GMT: 210.5 vs. 109.7, *p* = 0.162; Figure 2C) than those who were not infected by this strain. At time point E, at which time the Omicron XBB strain had spread, subjects infected with the Omicron or Omicron XBB strain exhibited higher NT_50_s against the XBB (GMT: 260.8 vs. 172.0, *p* = 0.482) and Omicron BA.2.86 strains (GMT: 288.4 vs. 193.1, *p* = 0.587) than their uninfected counterparts, although these differences were not significant.

### 3.3. Analysis of Cell-Mediated Immunity

The median number of SFCs (per 1 × 10^6^ PBMCs) at time points B, C, and E in the whole cohort were 78 [44–149], 208 [132–488], and 78 [45–450], respectively. The number of SFCs was significantly higher at time point C than at time points B (*p* < 0.001) and E (*p* = 0.014). A comparison of SFC counts in Omicron XBB-infected, Omicron-infected, and uninfected subjects is presented in Figure 3. The numbers of SFCs did not differ among these three groups of patients at any time point. At time point E, the median number of SFCs did not differ between subjects infected with the Omicron or Omicron XBB strain and uninfected subjects (82 vs. 78, *p* = 0.830).

### 3.4. Factors Affecting Humoral Immunity in the Subjects Received Multiple Vaccination

Multivariate linear regression analysis was performed to investigate the factors affecting NT_50_s against the Omicron BA.2.86 strain. At time point E, subjects’ mean age (1.080 [95% CI = −63.2 to 65.4]), sex (−475.2 [95% CI = −1556 to 605.6]), number of vaccinations (262.9 [95% CI = −150.9 to 676.6]), and SARS-CoV-2 infection history (456.6 [95% CI = −507.2 to 1420]). At time point E, receiver–operator analysis of NT50 against the XBB.1.5 and BA.2.86 strains yielded an area under the curve of 0.58 [95% CI = 0.37–0.78] (*p* = 0.47) for XBB.1.5 and 0.56 [95% CI = 0.35–0.76] (*p* = 0.57) for BA.2.86.

## 4. Discussion

This study analyzed humoral and cell-mediated immunity over a 4-year study period among healthcare personnel who received multiple mRNA vaccinations against COVID-19. Antibody titers against the original strain were maintained after the third vaccination. Antibodies against emerging strains such as Omicron XBB and Omicron BA.2.86 were observed in subjects with no exposure to these strains. Meanwhile, with additional vaccination optimized to the emerging strain (e.g., Omicron BA.4/5 and Omicron XBB strains), the antibody titer against the original strain did not decline. This result revealed that the vaccination against a specific strain introduced cross-immunity against the variant strains, in line with previous basic research [13]. Previous studies have indicated that repeated vaccination may lead to immune imprinting, whereby a robust immune response against the original strain may compromise the efficacy of new vaccines targeting emerging variants such as the Omicron XBB strain [14,15]. In contrast, Kotaki et al. demonstrated that updated vaccines shift antibody production from the original strain toward emerging variants, resulting in enhanced humoral responses to novel variants but diminished responses to the original strain [16]. This suggested that memory B cells primed with the original strain can be redirected from the original strain toward emerging Omicron variants. Conversely, Alsoussi et al. reported that SARS-CoV-2 booster immunizations do not redirect B cells biased toward the original strain but instead generate robust de novo B-cell responses primarily targeting variant-specific epitopes [17]. Notably, our present study revealed that the Omicron-directed updated vaccine elicited humoral immune responses against newly emerged variants without attenuating immunity against the original strain. These findings suggest the concurrent operation of both B-cell redirection and responses occurring in synergistic interplay. This immunological coordination is further supported by cohort evidence demonstrating that individuals who received three vaccinations against the original strain developed diverse antibodies exhibiting cross-reactivity against both the original strain and newly emerged variants [18]. Our results further substantiate the conclusion that multiple doses of Omicron-directed vaccines establish broad and effective humoral immunity against both emerging variants and the original strain through these dual mechanisms [19].

In the Omicron era, the receipt of three vaccine doses provided greater protection against SARS-CoV-2 infection than the receipt of two vaccine doses [20]. In our previous report, additional vaccine doses helped maintain or strengthen the antibody titers against circulating variant strains among subjects vaccinated 4–6 times [21]. We also reported that subjects with prior infection had approximately 2-fold higher spike-specific IgG titers than uninfected subjects [21]. The combination of natural infection and vaccination provided hybrid immunity [22]. However, among subjects who underwent multiple vaccinations (3–7 doses) with or without SARS-CoV-2 infection, the prior infection had no apparent effect on NT_50_s. In this study, the antibody titers of subjects who underwent multiple vaccinations appeared to converge, and the effect of natural infection on immunity became more difficult to recognize. These data were observed in study participants but not confirmed at the cellular level. A cell-level analysis of circulating peripheral mononuclear cells will be warranted to further characterize the specific B cell response to a particular endemic variant [23].

Newly emerged mutant strains, such as strain JN.1, a descendant of BA.2.86, have been reported to have high immune escape potential and a high propensity to cause reinfection in previously infected individuals [24]. In this study, the NT_50_s against XBB.1.5 and BA.2.86 were significantly lower than those against the original strain (D614G), supporting the evidence of the immune escape of the emerging strains. An antibody-dependent cell-mediated cytotoxicity assay using subjects’ sera may be a promising approach for future convalescent plasma therapy.

We evaluated cell-mediated immunity using the ELISpot analysis at three time points. The efficacy of the first two vaccinations in preventing infection declined after 6 months, but the efficacy in preventing severe disease was maintained longer [3]. Although the effect of vaccination on the peak SFC count was not clarified, the recent “human challenge study” revealed that after SARS-CoV-2 infection, the numbers of spike S1- and S2-specific SFCs (per 1 × 10^6^ PMBCs) increased from 141 to 409 [IQR 251–804] and from 211 to 467 [IQR 159–783], respectively, compared with those before the infection [25]. It has been reported that cell-mediated immunity also gradually declines after 6 months [26]. In this study, it was unclear why the number of SFCs was higher at time point D than at time points B and E. We consider that the initial two doses possibly provided a certain level of cellular immunity that was subsequently maintained [27]. T-cell activity induced by the vaccine also provided an immune response against the variants of concern [28].

The study had several limitations. The cohort was small, with approximately 10 subjects each in the Omicron XBB-infected, Omicron-infected, and uninfected groups. Our study also had potential selection bias, as the study participants were likely more sensitive to infections and more likely to undergo vaccination. This study included only immunocompetent individuals; therefore, our data may be inapplicable to immunosuppressed patients, such as those with cancer or organ transplants [29,30]. Ideally, the interval between blood sampling and the last vaccination should be standardized to more accurately assess the immune response at each time point. Herein, we used the IFN-γ ELISpot assay, which measures CD4^+^ and CD8^+^ T-cell responses. Analysis of CD4+ or CD8+ memory T-cells activity will be warranted to evaluate the role of T-cell in long-term immunity.

In this study, we focused on subjects infected with SARS-CoV-2. However, many RNA viruses mutate easily. Thus, vaccine development for RNA viruses is a key issue. Our study indicated that multiple vaccinations induced the production of antibodies with divergent activity against emerging mutant strains and enhanced protection the original strain. The difference in humoral immunity between the subjects with or without infection history was eliminated among subjects who underwent multiple vaccinations.

## Figures and Tables

**Figure 1 idr-17-00042-f001:**
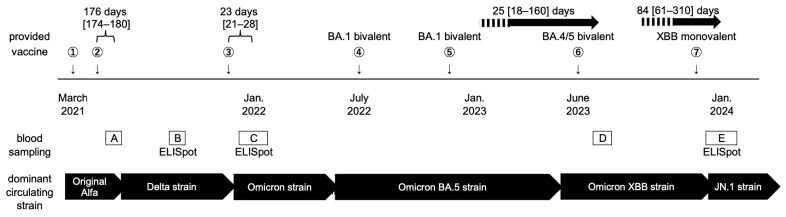
Schematic of this study.

**Figure 2 idr-17-00042-f002:**
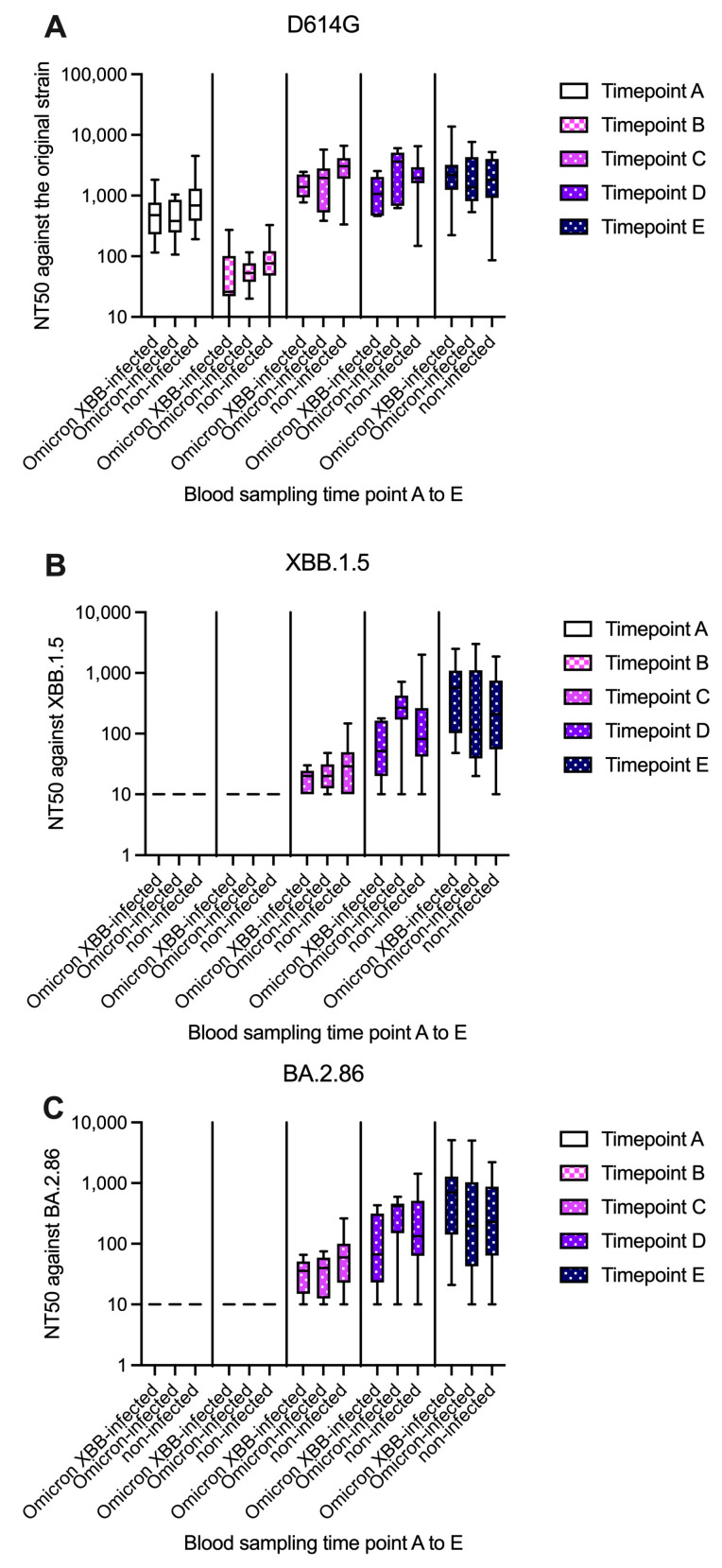
Neutralizing antibody titer (NT_50_) (**A**) against the original strain (D614G), (**B**) XBB.1.5 strain, and (**C**) BA.2.86 strain. The box-and-whisker plot presents the median, interquartile range, minimum, and maximum at time points A–E. Time points A and B were before the Omicron strain had emerged. At time point C, all subjects were uninfected. At time point D, nine subjects were infected by the Omicron strain. At time point E, an additional nine subjects were infected by the Omicron XBB strain. No subjects were infected twice.

**Figure 3 idr-17-00042-f003:**
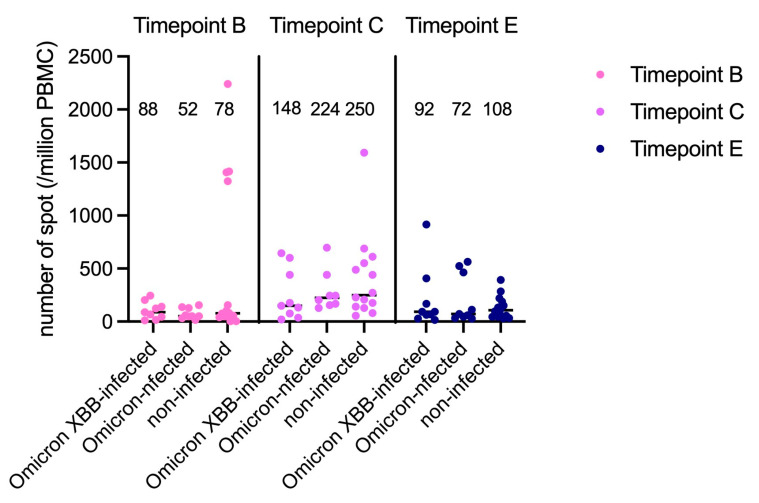
Specific lymphocyte counts as the number of spot-forming cells (SFCs) as determined by the ELISpot SARS-CoV-2 assay at time points B, C, and E. Each dot and bar represented an individual SFC count and the median.

## Data Availability

The data presented in this study are available on request from the corresponding author due to the future analysis of the data.

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
