# Peer review of "Humoral and Cell-Mediated Immunity Against SARS-CoV-2 in Healthcare Personnel Who Received Multiple mRNA Vaccines: A 4-Year Observational Study"

_2036-7449, 2025, doi:10.3390/idr17030042_

Round 1

Reviewer 1 Report

Comments and Suggestions for Authors

In the manuscript titled "Humoral and cell-mediated immunity against SARS-CoV-2 in healthcare personnel who received multiple mRNA vaccines: A 4-year observational study", the authors have investigated humoral and cellular immunity in 32 healthcare workers after vaccinations against SARS Cov-2. Some individuals in this study were later infected by different strains of SARS Cov-2 and the authors have compared their humoral and cellular immune responses with that of uninfected but vaccinated individuals. The authors showed that there were no notable differences between the infected and uninfected individuals in their humoral and cellular immune responses, highlighting the importance of repeated vaccinations.

Please address the following comments to revise the manuscript.

1) In figure 1, can the authors please add the detail regarding time between vaccination and the collection of sample? This has been mentioned in the results section and adding it in the figure will help with interpretation of the results.

2) Prior to figure 2, the authors can include a table for the vaccinations and which SARS Cov-2strains were those vaccines supposed to target? This will make understanding the subsequent figure easier.

3) Why did the authors choose D614G, XBB1.5, BA. 2.86 for the neutralization assay? If these were the most prevalent strains during a certain period, then please add that to the text for the benefit of the readers.

4) For figure 3, the authors mention that the spots were specific for a peptide pool of spike protein. Can the authors please provide more details about how this assay was performed? Was this peptide pool from a particular strain? Can this assay be done for peptides from spike proteins of different strains giving more specific readouts?

5) In line 228, the authors mention that "it is unclear why the numbers of SFCs was higher at time point D...", the authors should consider that this time point was closest the vaccination out of the 3 time points that were investigated.

6) The authors raise an important point in the discussion regarding the contributions of naive B cells or pre-existing memory B cells in the immune responses to subsequent SARS Cov-2 vaccinations or infections with new strains. Despite the role of memory B cells in secondary immune responses to SARS Cov-2, our assessment of humoral immunity is generally limited to antibody titers. In their study, did the authors consider investigating circulating memory B cells specific to different strains of SARS Cov-2?

Author Response

Thank you for the kind review. Here, we wrote the response to the reviewers’ comments. The corrected points in the manuscript were highlighted in red.

Reviewer 1

  1. In figure 1, can the authors please add the detail regarding time between vaccination and the collection of samples? This has been mentioned in the results section and adding it in the figure will help with interpretation of the results.

Thank you for your suggestion. We have added the period from the last vaccination to blood sampling points A–E in Figure 1.

2. Prior to figure 2, the authors can include a table for the vaccinations and which SARS Cov-2strains were those vaccines supposed to target? This will make understanding the subsequent figure easier.

Thank you for your suggestion. However, we are afraid we may not have completely understood the reviewer’s comment. Of course, we know that the strain used in the vaccine is most effective on its own, but it cross-reacts with other variants, so it is difficult to summarize in a table which vaccine targets which variants. Instead, we have included more detailed strain names together with vaccination history and blood sampling in Figure 1, which we hope will provide the reader with sufficient information.

3. Why did the authors choose D614G, XBB1.5, BA. 2.86 for the neutralization assay? If these were the most prevalent strains during a certain period, then please add that to the text for the benefit of the readers.

We selected XBB.1.5 and BA.2.86 strains as targets for the neutralizing titer analysis because these variants are widely predominant and exhibit strong immune escape capabilities. We have added a comment in lines 84–86 to clarify this.

4. For figure 3, the authors mention that the spots were specific for a peptide pool of spike protein. Can the authors please provide more details about how this assay was performed? Was this peptide pool from a particular strain? Can this assay be done for peptides from spike proteins of different strains giving more specific readouts?

We apologize for the lack of explanation regarding the ELISpot assay. The response to the variants of concern did not differ from that to the original strain. We have added relevant comments in the Methods section (lines 89–96).

5. In line 228, the authors mention that "it is unclear why the numbers of SFCs was higher at time point D...", the authors should consider that this time point was closest the vaccination out of the 3 time points that were investigated.

As the reviewer correctly highlighted, our study design for blood sampling timepoints was not correlated with additional vaccinations. This is a limitation of our study. We have added a statement to clarify this in the limitations (lines 258–259).

6) The authors raise an important point in the discussion regarding the contributions of naive B cells or pre-existing memory B cells in the immune responses to subsequent SARS Cov-2 vaccinations or infections with new strains. Despite the role of memory B cells in secondary immune responses to SARS Cov-2, our assessment of humoral immunity is generally limited to antibody titers. In their study, did the authors consider investigating circulating memory B cells specific to different strains of SARS Cov-2?

It is quite challenging to discriminate the circulating memory B cell population response to multiple viral strains. A detailed, cell-level analysis will be required to evaluate the response in circulating B cells. We have added a comment in the Discussion section (lines 231–233).

Reviewer 2 Report

Comments and Suggestions for Authors

An analysis of cellular immunity is needed, as CD4+ vs. CD8+ cells were not differentiated, nor was specific immunological memory assessed. Include phenotypic analysis of T cell subpopulations (flow cytometry).
Discuss the difference between infected and uninfected patients, as it is not statistically significant, but a relevant trend is suggested. Apply more sensitive analyses (ROC curves, mixed models) or clearly emphasize the lack of significance.
It would be interesting to add functional cross-neutralization assays using viral escape or ADCC assays here or in future research.
Does vaccination modify the humoral or cellular response to such an extent that it impacts the relationship between associated comorbidities? Does vaccination make infection more or less permissible in patients with comorbidities, according to your data?

Author Response

Thank you for the kind review. Here, we wrote the response to the reviewers’ comments. The corrected points in the manuscript were highlighted in red.

Reviewer 2

  • An analysis of cellular immunity is needed, as CD4+ vs. CD8+ cells were not differentiated, nor was specific immunological memory assessed. Include phenotypic analysis of T cell subpopulations (flow cytometry).

The ELISpot assay was used to evaluate antigen-specific lymphocyte counts via IFN-γ release in response to specific antigens. Therefore, the ELISpot assay measured spike protein-specific cellular immunity but did not distinguish between CD4+⁺ and CD8+ lymphocyte responses. We apologize for the lack of this information. Comments have been added to the Methods section (lines 89–95) and the Discussion (lines 260–261).

  • Discuss the difference between infected and uninfected patients, as it is not statistically significant, but a relevant trend is suggested. Apply more sensitive analyses (ROC curves, mixed models) or clearly emphasize the lack of significance.

Thank you for your valuable insights. We have added a ROC curve analysis to predict NT50 values for differentiating SARS-CoV-2 infection history. Corresponding comments were included in the Statistical Methods (lines 120–121) and Results sections (lines 186–189).

  • It would be interesting to add functional cross-neutralization assays using viral escape or ADCC assays here or in future research.

The reviewer’s suggestion is highly appreciated. An ADCC assay would be beneficial in developing convalescent plasma therapy. We have included this as a future perspective in the Discussion section (lines 238–239).

  • Does vaccination modify the humoral or cellular response to such an extent that it impacts the relationship between associated comorbidities? Does vaccination make infection more or less permissible in patients with comorbidities, according to your data?

We are sure that certain underlying conditions may influence humoral and cell-mediated responses in vaccine recipients. Herein, only immunocompetent subjects were analyzed. As the reviewer noted, it remains unclear whether our findings apply to immunosuppressed individuals. We have added a comment on this point in the Discussion section, with supporting citations (lines 256–257).

Round 2

Reviewer 2 Report

Comments and Suggestions for Authors

The paper is ready for publication